# Do Vision Transformers See Like Convolutional Neural Networks?

**Maithra Raghu**
Google Research, Brain Team
maithrar@gmail.com

**Thomas Unterthiner**
Google Research, Brain Team
unterthiner@google.com

**Simon Kornblith**
Google Research, Brain Team
kornblith@google.com

**Chiyuan Zhang**
Google Research, Brain Team
chiyuan@google.com

**Alexey Dosovitskiy**
Google Research, Brain Team
adosovitskiy@google.com

## Abstract

Convolutional neural networks (CNNs) have so far been the de-facto model for visual data. Recent work has shown that (Vision) Transformer models (ViT) can achieve comparable or even superior performance on image classification tasks. This raises a central question: *how are Vision Transformers solving these tasks*? Are they acting like convolutional networks, or learning entirely different visual representations? Analyzing the internal representation structure of ViTs and CNNs on image classification benchmarks, we find striking differences between the two architectures, such as ViT having more uniform representations across all layers. We explore how these differences arise, finding crucial roles played by self-attention, which enables early aggregation of global information, and ViT residual connections, which strongly propagate features from lower to higher layers. We study the ramifications for spatial localization, demonstrating ViTs successfully preserve input spatial information, with noticeable effects from different classification methods. Finally, we study the effect of (pretraining) dataset scale on intermediate features and transfer learning, and conclude with a discussion on connections to new architectures such as the MLP-Mixer.

## 1 Introduction

Over the past several years, the successes of deep learning on visual tasks has critically relied on convolutional neural networks [20, 16]. This is largely due to the powerful inductive bias of spatial equivariance encoded by convolutional layers, which have been key to learning general purpose visual representations for easy transfer and strong performance. Remarkably however, recent work has demonstrated that Transformer neural networks are capable of equal or superior performance on image classification tasks at large scale [14]. These Vision Transformers (ViT) operate almost *identically* to Transformers used in language [13], using self-attention, rather than convolution, to aggregate information across locations. This is in contrast with a large body of prior work, which has focused on more explicitly incorporating image-specific inductive biases [30, 9, 4]

This breakthrough highlights a fundamental question: *how* are Vision Transformers solving these image based tasks? Do they act like convolutions, learning the same inductive biases from scratch? Or are they developing novel task representations? What is the role of scale in learning these representations? And are there ramifications for downstream tasks? In this paper, we study these questions, uncovering key representational differences between ViTs and CNNs, the ways in which these difference arise, and effects on classification and transfer learning. Specifically, our contributions are:

35th Conference on Neural Information Processing Systems (NeurIPS 2021).

- We investigate the internal representation structure of ViTs and CNNs, finding striking differences between the two models, such as ViT having more uniform representations, with greater similarity between lower and higher layers.
- Analyzing how local/global spatial information is utilised, we find ViT incorporates more global information than ResNet at lower layers, leading to quantitatively different features.
- Nevertheless, we find that incorporating local information at lower layers remains vital, with large-scale pre-training data helping early attention layers learn to do this
- We study the uniform internal structure of ViT, finding that skip connections in ViT are even more influential than in ResNets, having strong effects on performance and representation similarity.
- Motivated by potential future uses in object detection, we examine how well input spatial information is preserved, finding connections between spatial localization and methods of classification.
- We study the effects of dataset scale on transfer learning, with a linear probes study revealing its importance for high quality intermediate representations.

## 2 Related Work

Developing non-convolutional neural networks to tackle computer vision tasks, particularly Transformer neural networks [44] has been an active area of research. Prior works have looked at *local* multiheaded self-attention, drawing from the structure of convolutional receptive fields [30, 36], directly combining CNNs with self-attention [4, 2, 46] or applying Transformers to smaller-size images [6, 9]. In comparison to these, the Vision Transformer [14] performs even less modification to the Transformer architecture, making it especially interesting to compare to CNNs. Since its development, there has also been very recent work analyzing aspects of ViT, particularly robustness [3, 31, 28] and effects of self-supervision [5, 7]. Other recent related work has looked at designing hybrid ViT-CNN models [49, 11], drawing on structural differences between the models. Comparison between Transformers and CNNs are also recently studied in the text domain [41].

Our work focuses on the representational structure of ViTs. To study ViT representations, we draw on techniques from neural network representation similarity, which allow the quantitative comparisons of representations within and across neural networks [17, 34, 26, 19]. These techniques have been very successful in providing insights on properties of different vision architectures [29, 22, 18], representation structure in language models [48, 25, 47, 21], dynamics of training methods [33, 24] and domain specific model behavior [27, 35, 38]. We also apply *linear probes* in our study, which has been shown to be useful to analyze the learned representations in both vision [1] and text [8, 32, 45] models.

## 3 Background and Experimental Setup

Our goal is to understand whether there are differences in the way ViTs represent and solve image tasks compared to CNNs. Based on the results of Dosovitskiy et al. [14], we take a representative set of CNN and ViT models — ResNet50x1, ResNet152x2, ViT-B/32, ViT-B/16, ViT-L/16 and ViT-H/14. Unless otherwise specified, models are trained on the JFT-300M dataset [40], although we also investigate models trained on the ImageNet ILSVRC 2012 dataset [12, 37] and standard transfer learning benchmarks [50, 14]. We use a variety of analysis methods to study the layer representations of these models, gaining many insights into how these models function. We provide further details of the experimental setting in Appendix A.

**Representation Similarity and CKA (Centered Kernel Alignment)**: Analyzing (hidden) layer representations of neural networks is challenging because their features are distributed across a large number of neurons. This distributed aspect also makes it difficult to meaningfully compare representations across neural networks. Centered kernel alignment (CKA) [17, 10] addresses these challenges, enabling quantitative comparisons of representations within and across networks. Specifically, CKA takes as input $\mathbf{X} \in \mathbb{R}^{m \times p_1}$ and $\mathbf{Y} \in \mathbb{R}^{m \times p_2}$ which are representations (activation matrices), of two layers, with $p_1$ and $p_2$ neurons respectively, evaluated on the same $m$ examples. Letting $\boldsymbol{K} = \boldsymbol{X}\boldsymbol{X}^\top$ and $\boldsymbol{L} = \boldsymbol{Y}\boldsymbol{Y}^\top$ denote the Gram matrices for the two layers (which measures the similarity of a pair of datapoints according to layer representations) CKA computes:

$$\text{CKA}(\boldsymbol{K}, \boldsymbol{L}) = \frac{\text{HSIC}(\boldsymbol{K}, \boldsymbol{L})}{\sqrt{\text{HSIC}(\boldsymbol{K}, \boldsymbol{K})\text{HSIC}(\boldsymbol{L}, \boldsymbol{L})}}, \tag{1}$$

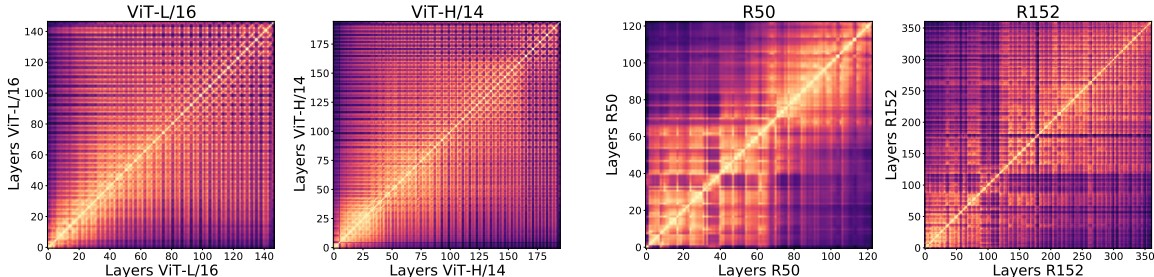

Figure 1: **Representation structure of ViTs and convolutional networks show significant differences, with ViTs having highly similar representations throughout the model, while the ResNet models show much lower similarity between lower and higher layers.** We plot CKA similarities between all pairs of layers across different model architectures. The results are shown as a heatmap, with the x and y axes indexing the layers from input to output. We observe that ViTs have relatively uniform layer similarity structure, with a clear grid-like pattern and large similarity between lower and higher layers. By contrast, the ResNet models show clear stages in similarity structure, with smaller similarity scores between lower and higher layers.

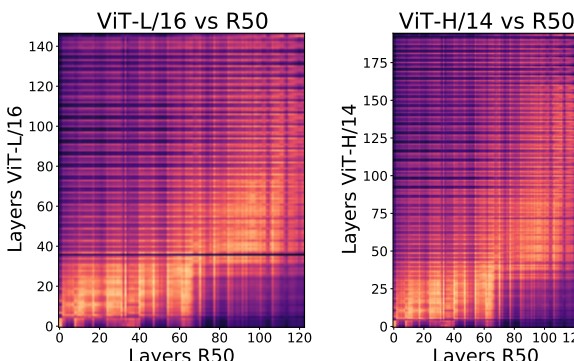

Figure 2: **Cross model CKA heatmap between ViT and ResNet illustrate that a larger number of lower layers in the ResNet are similar to a smaller set of the lowest ViT layers.** We compute a CKA heatmap comparing all layers of ViT to all layers of ResNet, for two different ViT models. We observe that the lower half of ResNet layers are similar to around the lowest quarter of ViT layers. The remaining half of the ResNet is similar to approximately the next third of ViT layers, with the highest ViT layers dissimilar to lower and higher ResNet layers.

where HSIC is the Hilbert-Schmidt independence criterion [15]. Given the centering matrix $\boldsymbol{H} = \boldsymbol{I}_n - \frac{1}{n}\boldsymbol{1}\boldsymbol{1}^\mathsf{T}$ and the centered Gram matrices $\boldsymbol{K}' = \boldsymbol{H}\boldsymbol{K}\boldsymbol{H}$ and $\boldsymbol{L}' = \boldsymbol{H}\boldsymbol{L}\boldsymbol{H}$, $\mathrm{HSIC}(\boldsymbol{K}, \boldsymbol{L}) = \mathrm{vec}(\boldsymbol{K}') \cdot \mathrm{vec}(\boldsymbol{L}')/(m-1)^2$, the similarity between these centered Gram matrices. CKA is invariant to orthogonal transformation of representations (including permutation of neurons), and the normalization term ensures invariance to isotropic scaling. These properties enable meaningful comparison and analysis of neural network hidden representations. To work at scale with our models and tasks, we approximate the unbiased estimator of HSIC [39] using minibatches, as suggested in [29].

## 4 Representation Structure of ViTs and Convolutional Networks

We begin our investigation by using CKA to study the internal representation structure of each model. How are representations propagated within the two architectures, and are there signs of functional differences? To answer these questions, we take every pair of layers $\boldsymbol{X}, \boldsymbol{Y}$ within a model and compute their CKA similarity. Note that we take representations not only from outputs of ViT/ResNet blocks, but also from intermediate layers, such as normalization layers and the hidden activations inside a ViT MLP. Figure 1 shows the results as a heatmap, for multiple ViTs and ResNets. We observe clear differences between the internal representation structure between the two model architectures: (1) ViTs show a much more uniform similarity structure, with a clear grid like structure (2) lower and higher layers in ViT show much greater similarity than in the ResNet, where similarity is divided into different (lower/higher) stages.

We also perform cross-model comparisons, where we take all layers $\boldsymbol{X}$ from ViT and compare to all layers $\boldsymbol{Y}$ from ResNet. We observe (Figure 2) that the lower half of 60 ResNet layers are similar to approximately the lowest quarter of ViT layers. In particular, many more lower layers in the ResNet are needed to compute similar representations to the lower layers of ViT. The top half of the ResNet is approximately similar to the next third of the ViT layers. The final third of ViT layers is less similar to

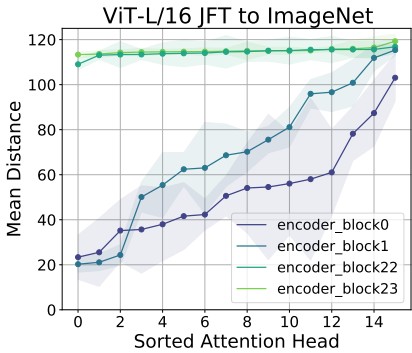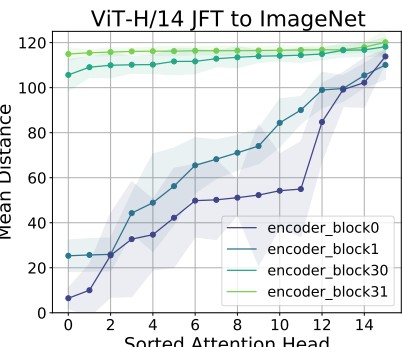

**Figure 3: Plotting attention head mean distances shows lower ViT layers attend both locally and globally, while higher layers primarily incorporate global information.** For each attention head, we compute the pixel distance it attends to, weighted by the attention weights, and then average over 5000 datapoints to get an average attention head distance. We plot the heads sorted by their average attention distance for the two lowest and two highest layers in the ViT, observing that the lower layers attend both locally and globally, while the higher layers attend entirely globally.

all ResNet layers, likely because this set of layers mainly manipulates the CLS token representation, further studied in Section 6.

Taken together, these results suggest that (i) ViT lower layers compute representations in a different way to lower layers in the ResNet, (ii) ViT also more strongly propagates representations between lower and higher layers (iii) the highest layers of ViT have quite different representations to ResNet.

## 5 Local and Global Information in Layer Representations

In the previous section, we observed much greater similarity between lower and higher layers in ViT, and we also saw that ResNet required more lower layers to compute similar representations to a smaller set of ViT lower layers. In this section, we explore one possible reason for this difference: the difference in the ability to incorporate global information between the two models. How much global information is aggregated by early self-attention layers in ViT? Are there noticeable resulting differences to the features of CNNs, which have fixed, local receptive fields in early layers? In studying these questions, we demonstrate the influence of global representations and a surprising connection between scale and self-attention distances.

**Analyzing Attention Distances:** We start by analyzing ViT self-attention layers, which are the mechanism for ViT to aggregate information from other spatial locations, and structurally very different to the fixed receptive field sizes of CNNs. Each self-attention layer comprises multiple self-attention heads, and for each head we can compute the average distance between the query patch position and the locations it attends to. This reveals how much local vs global information each self-attention layer is aggregating for the representation. Specifically, we weight the pixel distances by the attention weights for each attention head and average over 5000 datapoints, with results shown in Figure 3. In agreement with Dosovitskiy et al. [14], we observe that even in the lowest layers of ViT, self-attention layers have a mix of local heads (small distances) and global heads (large distances). This is in contrast to CNNs, which are hardcoded to attend only locally in the lower layers. At higher layers, all self-attention heads are global.

Interestingly, we see a clear effect of scale on attention. In Figure 4, we look at attention distances when training *only* on ImageNet (no large-scale pre-training), which leads to much lower performance in ViT-L/16 and ViT-H/14 [14]. Comparing to Figure 3, we see that with not enough data, ViT *does not learn to attend locally* in earlier layers. Together, this suggests that using local information early on for image tasks (which is hardcoded into CNN architectures) is important for strong performance.

**Does access to global information result in different features?** The results of Figure 3 demonstrate that ViTs have access to more global information than CNNs in their lower layers. But does this result in different learned features? As an interventional test, we take subsets of the ViT attention heads from the first encoder block, ranging from the subset corresponding to the most local attention

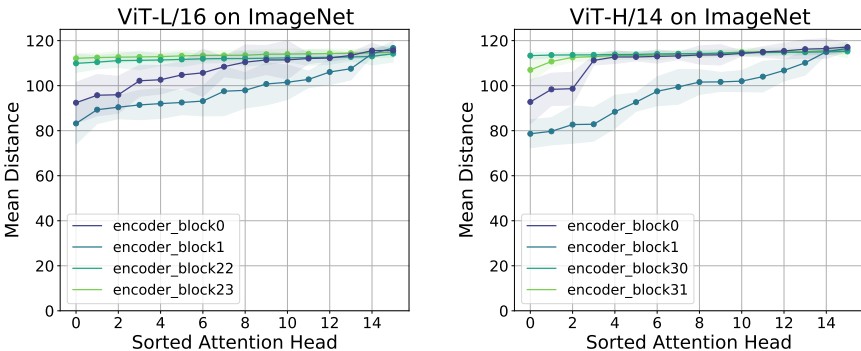

**Figure 4: With less training data, lower attention layers do not learn to attend locally.** Comparing the results to Figure 3, we see that training only on ImageNet leads to the lower layers not learning to attend more locally. These models also perform much worse when only trained on ImageNet, suggesting that incorporating local features (which is hardcoded into CNNs) may be important for strong performance. (See also Figure C.5.)

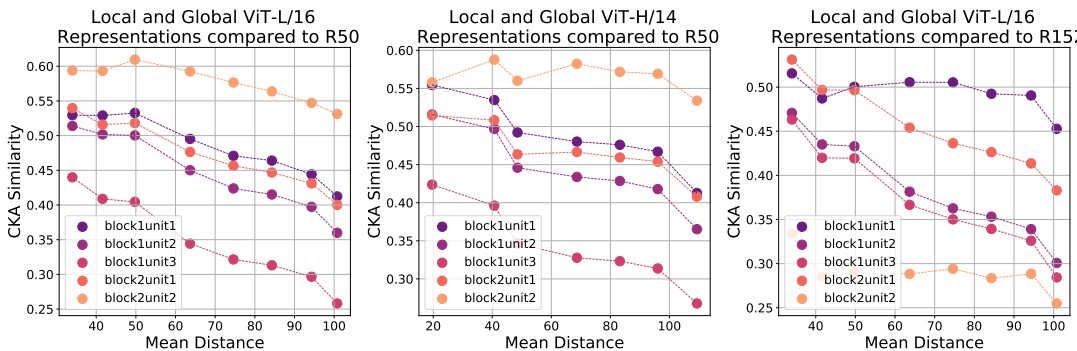

**Figure 5: Lower layer representations of ResNet are most similar to representations corresponding to local attention heads of ViT.** We take subsets of ViT attention heads in the first encoder block, ranging from the most locally attending heads (smallest mean distance) to the most global heads (largest mean distance). We then compute CKA similarity between these subsets and lower layer representations in the ResNet. We observe that lower ResNet layers are most similar to the features learned by local attention heads of ViT, and decrease monotonically in similarity as more global information is incorporated, demonstrating that the global heads learn quantitatively different features.

heads to a subset of the representation corresponding to the most global attention heads. We then compute CKA similarity between these subsets and the lower layer representations of ResNet.

The results, shown in Figure 5, which plot the mean distance for each subset against CKA similarity, clearly show a monotonic decrease in similarity as mean attention distance grows, demonstrating that access to more global information also leads to quantitatively different features than computed by the local receptive fields in the lower layers of the ResNet.

**Effective Receptive Fields:** We conclude by computing *effective receptive fields* [23] for both ResNets and ViTs, with results in Figure 6 and Appendix C. We observe that lower layer effective receptive fields for ViT are indeed larger than in ResNets, and while ResNet effective receptive fields grow gradually, ViT receptive fields become much more global midway through the network. ViT receptive fields also show strong dependence on their center patch due to their strong residual connections, studied in the next section. As we show in Appendix C, in attention sublayers, receptive fields taken before the residual connection show far less dependence on this central patch.

## 6 Representation Propagation through Skip Connections

The results of the previous section demonstrate that ViTs learn different representations to ResNets in lower layers due to access to global information, which explains some of the differences in represen-

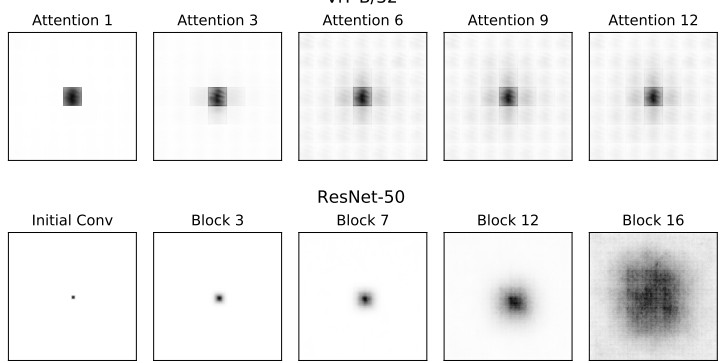

Figure 6: **ResNet effective receptive fields are highly local and grow gradually; ViT effective receptive fields shift from local to global**. We measure the effective receptive field of different layers as the absolute value of the gradient of the center location of the feature map (taken after residual connections) with respect to the input. Results are averaged across all channels in each map for 32 randomly-selected images.

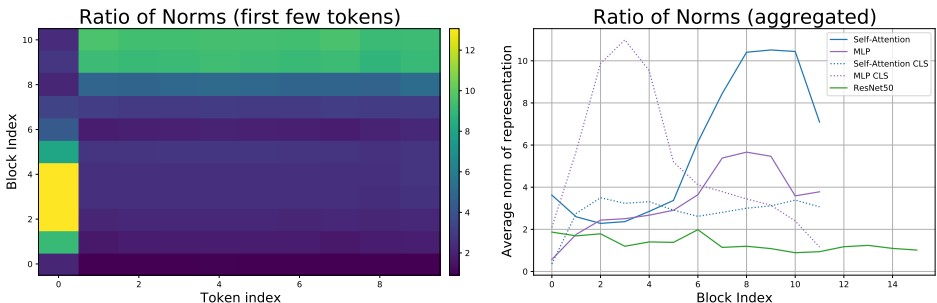

Figure 7: **Most information in ViT passes through skip connections**. Comparison of representation norms between the skip-connection (identity) and the long branch for ViT-B/16 trained on ImageNet and a ResNet. For ViT, we show the CLS token separately from the rest of the representation. (left) shows the ratios separated for the first few tokens (token 0 is CLS), (right) shows averages over all tokens.

tation structure observed in Section 4. However, the highly uniform nature of ViT representations (Figure 1) also suggests lower representations are faithfully propagated to higher layers. But how does this happen? In this section, we explore the role of skip connections in representation propagation across ViTs and ResNets, discovering ViT skip connections are highly influential, with a clear phase transition from preserving the CLS (class) token representation (in lower layers) to spatial token representations (in higher layers).

Like Transformers, ViTs contain *skip* (aka *identity* or *shortcut*) connections throughout, which are added on after the (i) self-attention layer, and (ii) MLP layer. To study their effect, we plot the norm ratio $||z_i||/||f(z_i)||$ where $z_i$ is the hidden representation of the $i$th layer coming from the skip connection, and $f(z_i)$ is the transformation of $z_i$ from the *long branch* (i.e. MLP or self-attention.)

The results are in Figure 7 (with additional cosine similarity analysis in Figure E.2.) The heatmap on the left shows $||z_i||/||f(z_i)||$ for different token representations. We observe a striking phase transition: in the first half of the network, the CLS token (token 0) representation is primarily propagated by the skip connection branch (high norm ratio), while the spatial token representations have a large contribution coming from the long branch (lower norm ratio). Strikingly, in the second half of the network, this is reversed.

The right pane, which has line plots of these norm ratios across ResNet50, the ViT CLS token and the ViT spatial tokens additionally demonstrates that skip connection is much more influential in ViT compared to ResNet: we observe much higher norm ratios for ViT throughout, along with the phase transition from CLS to spatial token propagation (shown for the MLP and self-attention layers.)

**ViT Representation Structure without Skip Connections:** The norm ratio results strongly suggest that skip connections play a key role in the representational structure of ViT. To test this interventionally, we train ViT models with skip connections removed in block $i$ for varying $i$, and plot the CKA representation heatmap. The results, in Figure 8, illustrate that removing the skip connections in a block partitions the layer representations on either side. (We note a performance drop of $4\%$ when

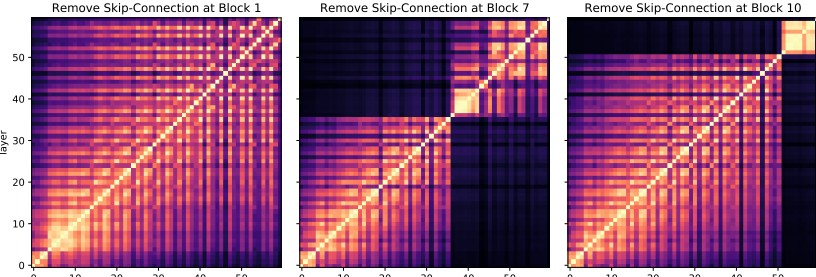

**Figure 8: ViT models trained without any skip connections in block $i$ show very little representation similarity between layers before/after block $i$.** We train several ViT models without any skip connections at block $i$ for varying $i$ to interventionally test the effect on representation structure. For middle blocks without skip connections, we observe a performance drop of $4\%$. We also observe that removing a skip connection at block $i$ partitions similar representations to before/after block $i$ — this demonstrates the importance of skip connections in ViT's standard uniform representation structure.

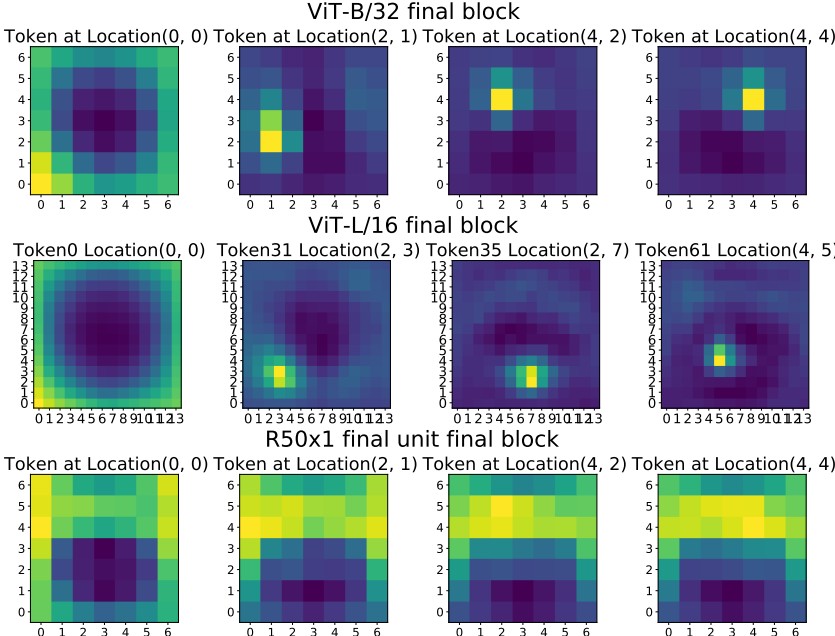

**Figure 9: Higher layers of ViT maintain spatial location information more faithfully than ResNets**. Each heatmap plot shows the CKA similarity between a single token representation in final block of the model and the input images, which are divided into non-overlapping patches. We observe that ViT tokens have strongest similarity to their corresponding spatial location in the image, but tokens corresponding to spatial locations at the edge of the image (e.g. token 0) additionally show similarity to other edge positions. This demonstrates that spatial information from the input is preserved even at the final layer of ViT. By contrast, ResNet "tokens" (features at a specific spatial location) are much less spatially discriminative, showing comparable similarity across a broad set of input spatial locations. See Appendix for additional layers and results.

removing skip connections from middle blocks.) This demonstrates the importance of representations being propagated by skip connections for the uniform similarity structure of ViT in Figure 1.

# 7 Spatial Information and Localization

The results so far, on the role of self-attention in aggregating spatial information in ViTs, and skip-connections faithfully propagating representations to higher layers, suggest an important followup question: how well can ViTs perform *spatial localization*? Specifically, is spatial information from the input preserved in the higher layers of ViT? And how does it compare in this aspect to ResNet? An affirmative answer to this is crucial for uses of ViT beyond classification, such as object detection.

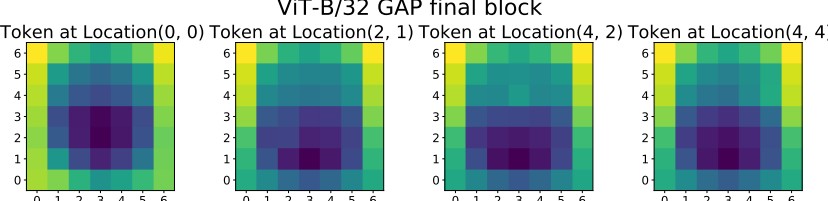

ViT-B/32 GAP final block

**Figure 10: When trained with global average pooling (GAP) instead of a CLS token, ViTs show less clear localization (compare Figure 9).** We plot the same CKA heatmap between a token and different input images patches as in Figure 9, but for a ViT model trained with global average pooling (like ResNet) instead of a CLS token. We observe significantly less localization.

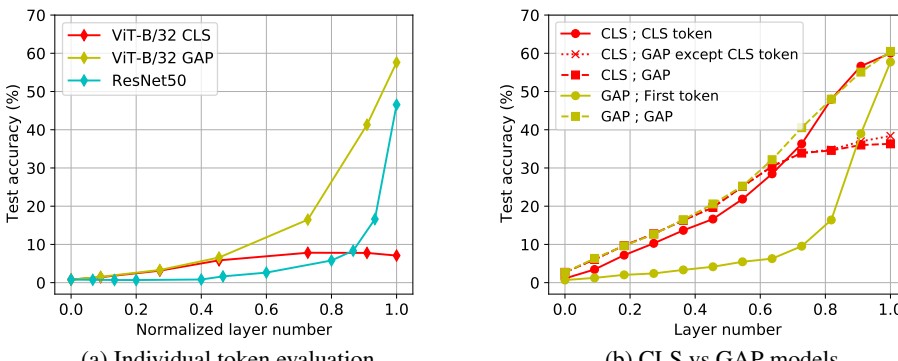

(a) Individual token evaluation

(b) CLS vs GAP models

**Figure 11: Spatial localization experiments with linear probes.** We train linear classifiers on 10-shot ImageNet classification from the representations extracted from different layers of ViT-B/32 models. We then plot the accuracy of the probe versus the (normalized) layer number. **Left:** We train a classifier on each token separately and report the average accuracy over all tokens (excluding the CLS token for the ViT CLS model.) **Right:** Comparison of ViT models pre-trained with a classification token or with global average pooling (GAP) and then evaluated with different ways of aggregating the token representations.

We begin by comparing token representations in the higher layers of ViT and ResNet to those of input patches. Recall that ViT tokens have a corresponding input patch, and thus a corresponding input spatial location. For ResNet, we define a token representation to be all the convolutional channels at a particular spatial location. This also gives it a corresponding input spatial location. We can then take a token representation and compute its CKA score with input image patches at different locations. The results are illustrated for different tokens (with their spatial locations labelled) in Figure 9.

For ViT, we observe that tokens corresponding to locations at the edge of the image are similar to edge image patches, but tokens corresponding to interior locations are well localized, with their representations being most similar to the corresponding image patch. By contrast, for ResNet, we see significantly weaker localization (though Figure D.3 shows improvements for earlier layers.)

One factor influencing this clear difference between architectures is that ResNet is trained to classify with a global average pooling step, while ViT has a separate classification (CLS) token. To examine this further, we test a ViT architecture trained with global average pooling (GAP) for localization (see Appendix A for training details). The results, shown in Figure 10, demonstrate that global average pooling does indeed reduce localization in the higher layers. More results in Appendix Section D.

**Localization and Linear Probe Classification:** The previous results have looked at localization through direct comparison of each token with input patches. To complete the picture, we look at using each token separately to perform *classification* with linear probes. We do this across different layers of the model, training linear probes to classify image label with closed-form few-shot linear regression similar to Dosovitskiy et al. [14] (details in Appendix A). Results are in Figure 11, with further results in Appendix F. The left pane shows average accuracy of classifiers trained on individual tokens, where we see that ResNet50 and ViT with GAP model tokens perform well at higher layers, while in the standard ViT trained with a CLS token the spatial tokens do poorly – likely because their representations remain spatially localized at higher layers, which makes global classification challenging. Supporting this are results on the right pane, which shows that a single token from the

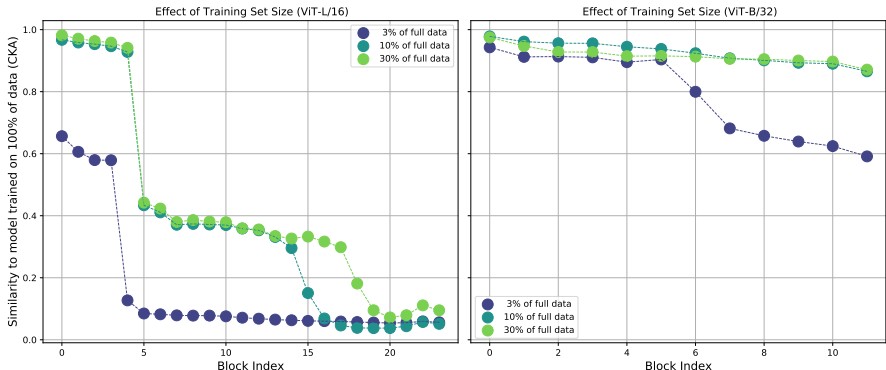

**Figure 12: Measuring similarity of representations learned with varying amounts of data shows the importance of large datasets for higher layers and larger model representations.** We compute the similarity of representations at each block for ViT models that have been trained on smaller subsets of the data to a model that has been trained on the full data on ViT-L/16 (left) and ViT-B/32 (right). We observe that while lower layer representations have high similarity even with 10% of the data, higher layers and larger models require significantly more data to learn similar representations.

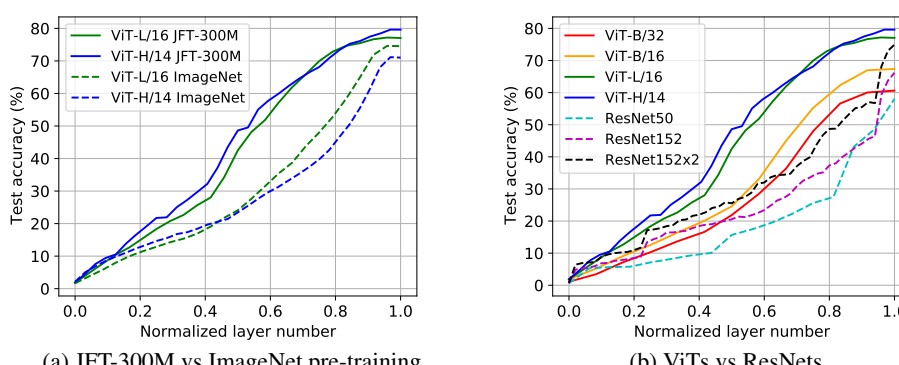

(a) JFT-300M vs ImageNet pre-training  (b) ViTs vs ResNets

**Figure 13:** Experiments with linear probes. We train linear classifiers on 10-shot ImageNet classification from the aggregated representations of different layers of different models. We then plot the accuracy of the probe versus the (normalized) layer number. **Left:** Comparison of ViTs pre-trained on JFT-300M or ImageNet and evaluated with linear probes on Imagenet. **Right:** Comparison of ViT and ResNet models trained JFT-300m, evaluated with linear probes on ImageNet.

ViT-GAP model achieves comparable accuracy in the highest layer to all tokens pooled together. With the results of Figure 9, this suggests all higher layer tokens in GAP models learn similar (global) representations.

## 8   Effects of Scale on Transfer Learning

Motivated by the results of Dosovitskiy et al. [14] that demonstrate the importance of dataset scale for high performing ViTs, and our earlier result (Figure 4) on needing scale for local attention, we perform a study of the effect of dataset scale on representations in transfer learning.

We begin by studying the effect on representations as the JFT-300M pretraining dataset size is varied. Figure 12 illustrates the results on ViT-B/32 and ViT-L/16. Even with 3% of the entire dataset, lower layer representations are very similar to the model trained on the whole dataset, but higher layers require larger amounts of pretraining data to learn the same representations as at large data scale, especially with the large model size. In Section G, we study how much representations change in finetuning, finding heterogeneity over datasets.

We next look at dataset size effect on the larger ViT-L/16 and ViT-H/14 models. Specifically, in the left pane of Figure 13, we train linear classifer probes on ImageNet classes for models pretrained on JFT-300M vs models only pretrained on ImageNet. We observe the JFT-300M pretained models achieve much higher accuracies even with middle layer representations, with a 30% gap in absolute

accuracy to the models pretrained only on ImageNet. This suggests that for larger models, the larger dataset is especially helpful in learning high quality intermediate representations. This conclusion is further supported by the results of the right pane of Figure 13, which shows linear probes on different ResNet and ViT models, all pretrained on JFT-300M. We again see the larger ViT models learn much stronger intermediate representations than the ResNets. Additional linear probes experiments in Section F demonstrate this same conclusion for transfer to CIFAR-10 and CIFAR-100.

## 9    Discussion

**Limitations:** Our study uses CKA [17], which summarizes measurements into a single scalar, to provide quantitative insights on representation similarity. While we have complemented this with interventional tests and other analyses (e.g. linear probes), more fine-grained methods may reveal additional insights and variations in the representations.

**Conclusion:** Given the central role of convolutional neural networks in computer vision break-throughs, it is remarkable that Transformer architectures (almost *identical* to those used in language) are capable of similar performance. This raises fundamental questions on whether these architectures work in the same way as CNNs. Drawing on representational similarity techniques, we find surprisingly clear differences in the features and internal structures of ViTs and CNNs. An analysis of self-attention and the strength of skip connections demonstrates the role of earlier global features and strong representation propagation in ViTs for these differences, while also revealing that some CNN properties, e.g. local information aggregation at lower layers, are important to ViTs, being learned from scratch at scale. We examine the potential for ViTs to be used beyond classification through a study of spatial localization, discovering ViTs with CLS tokens show strong preservation of spatial information — promising for future uses in object detection. Finally, we investigate the effect of scale for transfer learning, finding larger ViT models develop significantly stronger intermediate representations through larger pretraining datasets. These results are also very pertinent to under-standing MLP-based architectures for vision proposed by concurrent work [42, 43], further discussed in Section H, and together answer central questions on differences between ViTs and CNNs, and suggest new directions for future study. From the perspective of societal impact, these findings and future work may help identify potential failures as well as greater model interpretability.

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
