# Appendix

Additional details and results from the different sections are included below.

## A    Additional details on Methods and the Experimental Setup

To understand systematic differences between ViT and CNNs, we use a representative set of different models of each type, guided by the performance results in [14]. Specifically, for ViTs, we look at ViT-B/32, ViT-L/16 and ViT-H/14, where the smallest model (ViT-B/32) shows limited improvements when pretraining on JFT-300M [40] vs. the ImageNet Large Scale Visual Recognition Challenge 2012 dataset [12, 37], while the largest, ViT-H/14, achieves state of the art when pretrained on JFT-300M [40]. ViT-L/16 is close to the performance ViT-H/14 [14]. For CNNs, we look at ResNet50x1 which also shows saturating performance when pretraining on JFT-300M, and also ResNet152x2, which in contrast shows large performance gains with increased pretraining dataset size. As in Dosovitskiy et al. [14], these ResNets follow some of the implementation changes first proposed in BiT [16].

In addition to the standard Vision Transformers trained with a classification token (CLS), we also trained ViTs with global average pooling (GAP). In these, there is no classification token – instead, the representations of tokens in the last layer of the transformer are averaged and directly used to predict the logits. The GAP model is trained with the same hyperparameters as the CLS one, except for the initial learning rate that is set to a lower value of $0.0003$.

For analyses of internal model representations, we observed no meaningful difference between representations of images drawn from ImageNet and images drawn from JFT-300M. Figures 1, 2, 9, and 10 use images from the JFT-300M dataset that were not seen during training, while Figures 6, 3, 4, 5 7, 8, and 12 use images from the ImageNet 2012 validation set. Figures 11 and 13 involve 10-shot probes trained on the ImageNet 2012 training set, tuned hyperparameters on a heldout portion of the training set, and evaluated on the validation set.

**Additional details on CKA implementation**    To compute CKA similarity scores, we use minibatch CKA, introduced in [29]. Specifically, we use batch sizes of $1024$ and we sample a total of $10240$ examples without replacement. We repeat this $20$ times and take the average. Experiments varying the exact batch size (down to $128$), and total number of examples used for CKA ( down to $2560$ total examples, repeated $10$ times), had no noticeable effect on the results.

**Additional details on linear probes.**    We train linear probes as regularized least-squares regression, following Dosovitskiy et al. [14]. We map the representations training images to $\{-1, 1\}^N$ target vectors, where $N$ is the number of classes. The solution can be recovered efficiently in closed form.

For vision transformers, we train linear probes on representations from individual tokens or on the representation averaged over all tokens, at the output of different transformer layers (each layer meaning a full transformer block including self-attention and MLP). For ResNets, we take representation at the output of each residual block (including 3 convolutional layers). The resolution of the feature maps changes throughout the model, so we perform an additional pooling step bringing the feature map to the same spatial size as in the last stage. Moreover, ResNets differ from ViTs in that the number of channels changes throughout the model, with fewer channels in the earlier layers. This smaller channel count in the earlier layers could potentially lead to worse performance of the linear probes. To compensate for this, before pooling we split the feature map into patches and flattened each patch, so as to arrive at the channel count close to the channel count in the final block. All results presented in the paper include this additional patching step; however, we have found that it brings only a very minor improvement on top of simple pooling.

## B    Additional Representation Structure Results

Here we include some more CKA heatmaps, which provide insights on model representation structures (compare to Figure 1, Figure 2 in the main text.) We observe similar conclusions: ViT representation structure has a more uniform similarity structure across layers, and comparing ResNet to ViT

representations show a large fraction of early ResNet layers similar to a smaller number of ViT layers.

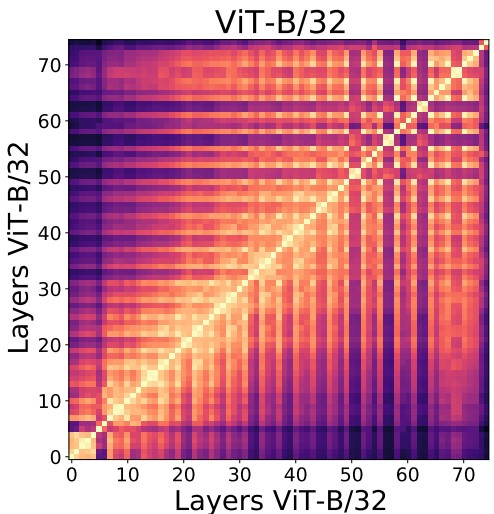

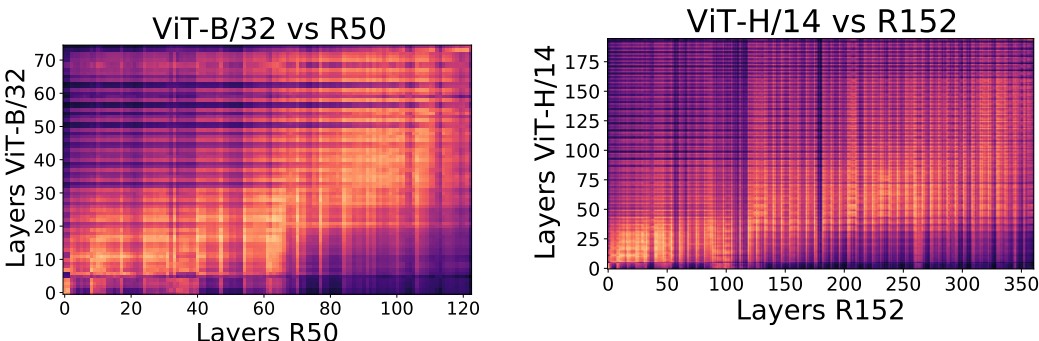

**Figure B.1: Additional CKA heatmap results.** Top shows CKA heatmap for ViT-B/32, where we can also observe strong similarity between lower and higher layers and the grid like, uniform representation structure. Bottom shows (i) ViT-B/32 compared to R50, where we again see that 60 of the lowest R50 layers are similar to about 25 of the lowest ViT-B/32 layers, with the remaining layers most similar to each other (ii) ViT-H/14 compared to R152, where we see the lowest 100 layers of R152 are most similar to lowest 30 layers of ViT-H/14.

# C   Additional Local/Global Information Results

In Figures C.1, C.2, and C.3, we provide full plots of effective receptive fields of all layers of ViT-B/32, ResNet-50, and ViT-L/16, taken after the residual connections as in Figure 6 in the text. In Figure C.4, we show receptive fields of ViT-B/32 and ResNet-50 taken *before* the residual connections. Although the pre-residual receptive fields of ViT MLP sublayers resemble the post-residual receptive fields in Figure C.1, the pre-residual receptive fields of attention sublayers have a smaller relative contribution from the corresponding input patch. These results support our findings in Section 5 regarding the global nature of attention heads, but suggest that network representations remain tied to input patch locations because of the strong contributions from skip connections, studied in Section 6. ResNet-50 pre-residual receptive fields look similar to the post-residual receptive fields.

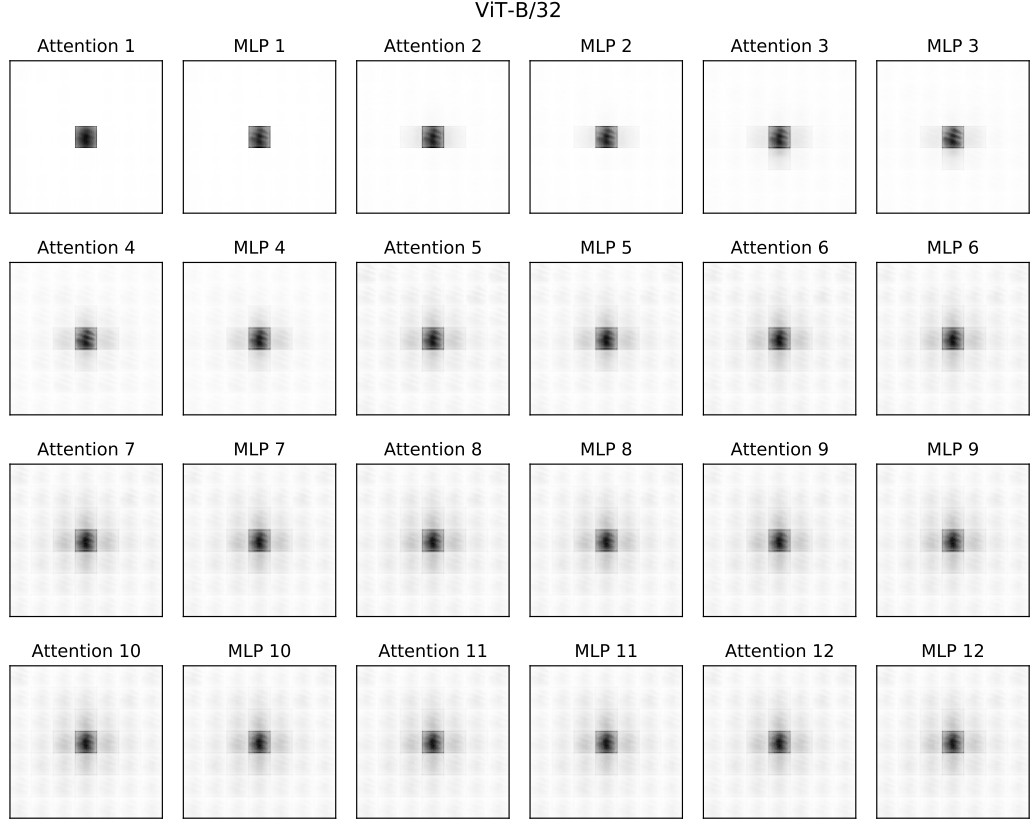

**Figure C.1: Post-residual receptive fields of all ViT-B/32 sublayers.**

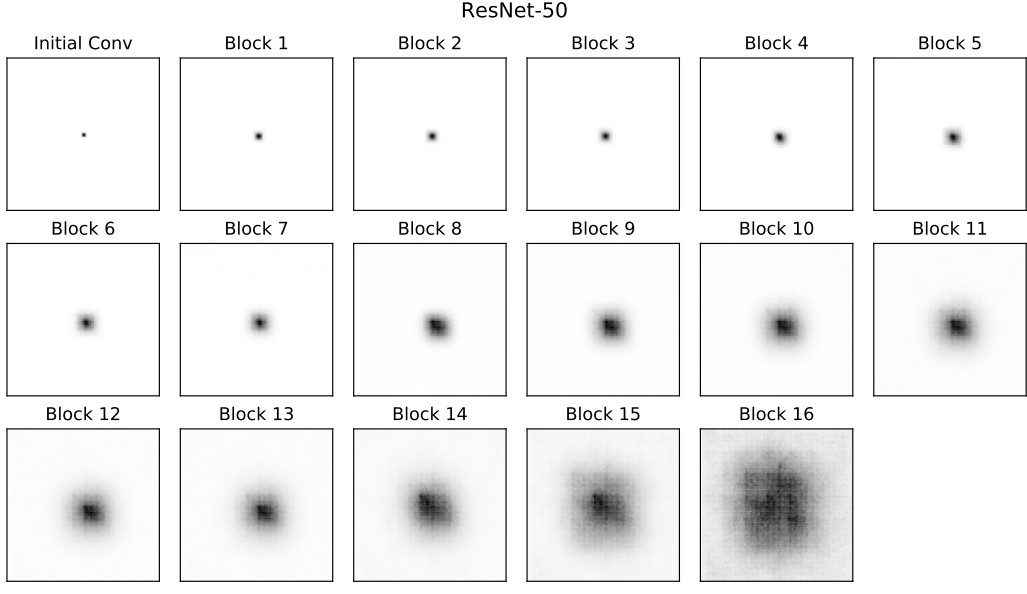

**Figure C.2: Post-residual receptive fields of all ResNet-50 blocks.**

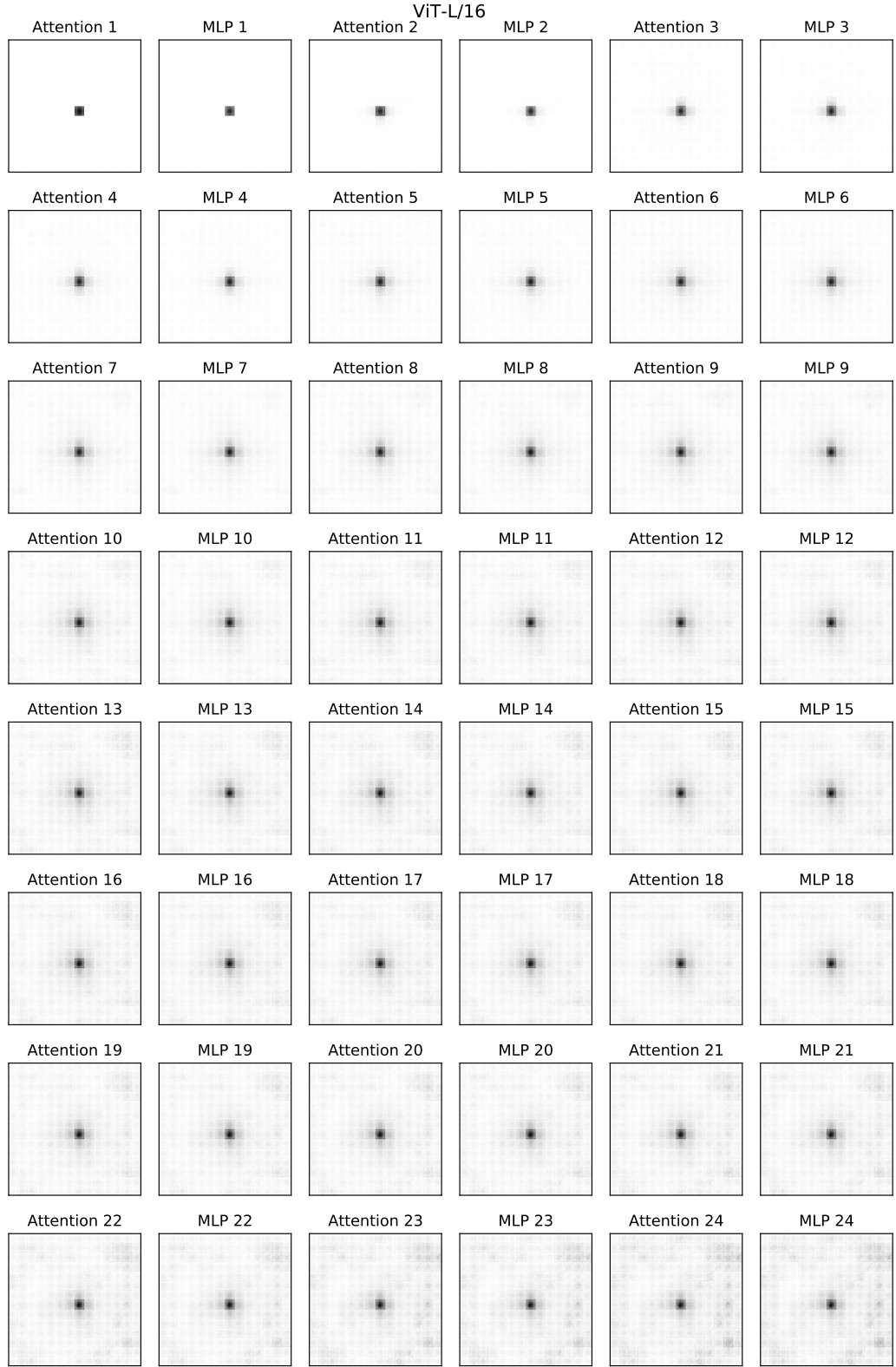

**Figure C.3: Post-residual receptive fields of all ViT-L/16 sublayers.**

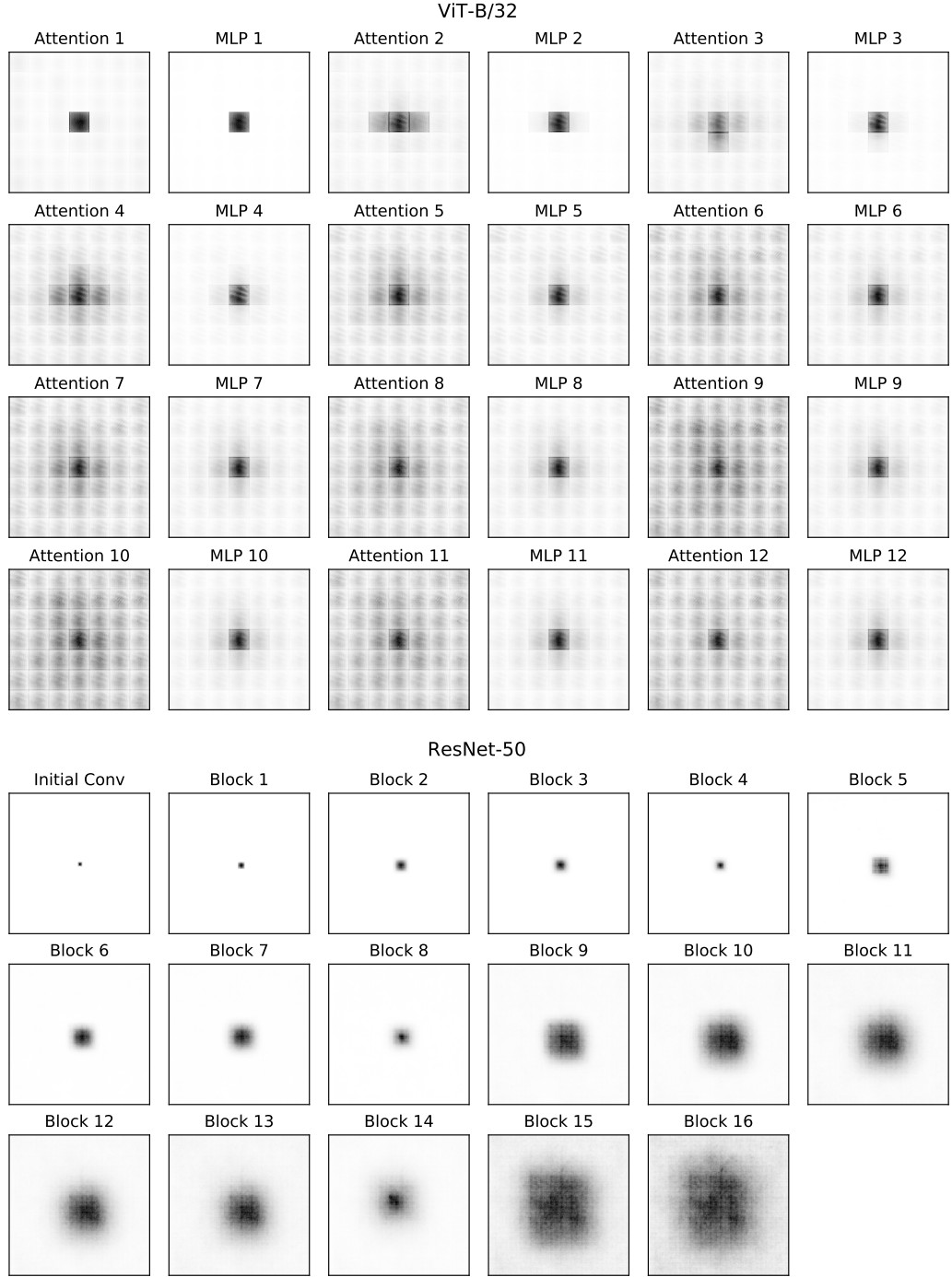

**Figure C.4: Pre-residual receptive fields of all ViT-B/32 sublayers and ResNet-50 blocks.** In ViT-B, we see that the pre-residual receptive fields of later attention sublayers are not dominated by the center patch, in contrast to the post-residual receptive fields shown in Figure C.1. Thus, although later attention sublayers integrate information across the entire input image, network representations remain localized due to the strong skip connections. ResNet-50 pre-residual receptive fields generally resemble the post-residual receptive fields shown in Figure C.2. The receptive field appears to "shrink" at blocks 4, 8, and 14, which are each the first in a stage, and for which we plot only the longer branch and not the shortcut. The receptive field does not shrink when computed after the summation of these branches, as shown in Figure C.2.

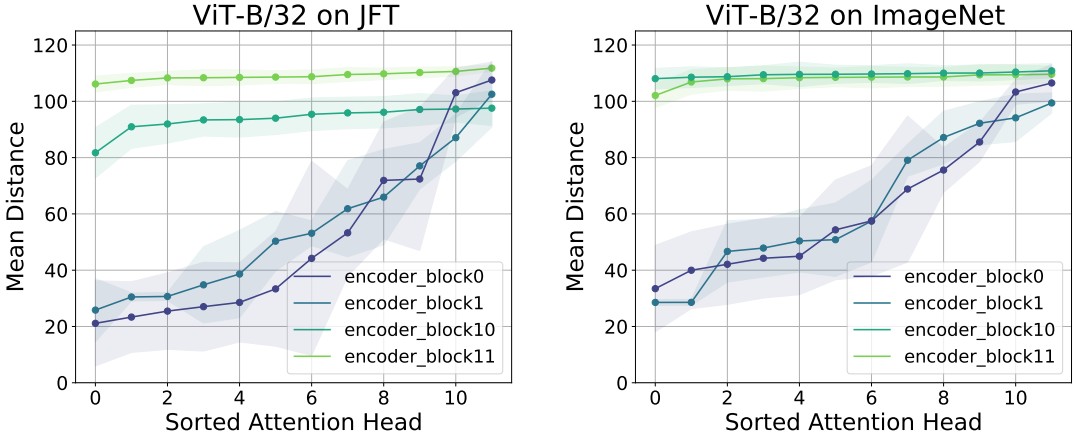

**Figure C.5: Plot of attention head distances for ViT-B/32 when trained on JFT-300M and on only ImageNet shows that ViT-B/32 learns to attend locally even on a smaller dataset.** Compare to Figure 3 in the main text. While ViT-L and ViT-H show large performance improvements when (i) finetuned on ImageNet having been pretrained on JFT compared to (ii) being trained on only ImageNet, ViT-B/32 has similar performance in both settings. We also observe that ViT-H, ViT-L don't learn to attend locally in the lowest layers when only trained on ImageNet (Figure 3), whereas here we see ViT-B/32 still learns to attend locally — suggesting connections between performance and heads learning to attend locally.

# D  Localization

Below we include additional localization results: computing CKA between different input patches and tokens in the higher layers of the models. We show results for ViT-H/14, additional higher layers for ViT-L/16, ViT-B/32 and addtional layers for ResNet.

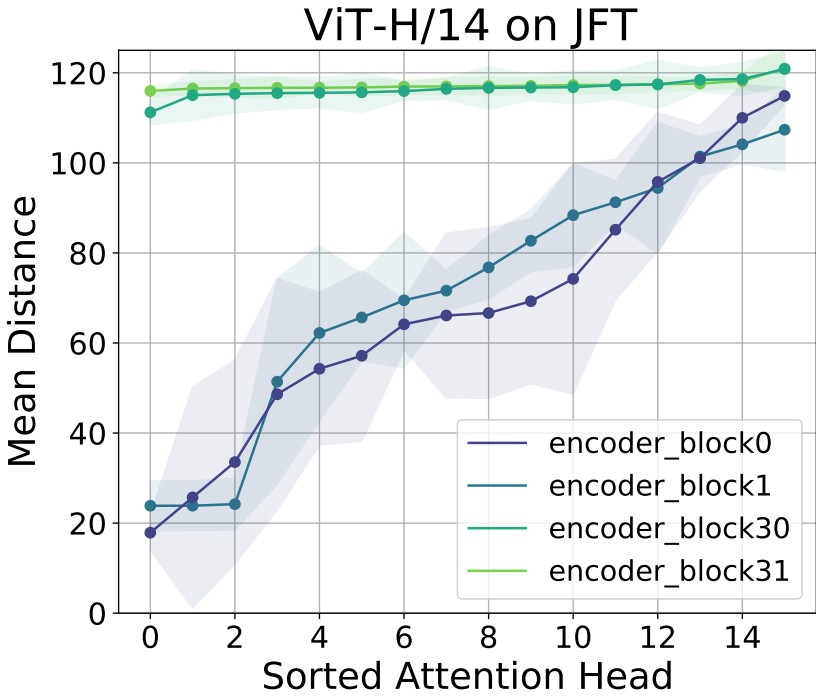

**Figure C.6: Additional plot of attention head distances for ViT-H/14 on JFT-300M.** Compare to Figure 3 in the main text. For each attention head, we compute the pixel distance it attends to, weighted by the attention weights, and then average over 5000 datapoints to get an average attention head distance. We plot the heads sorted by their average attention distance for the two lowest and two highest layers in the ViT, observing that the lower layers attend both locally and globally, while the higher layers attend entirely globally.

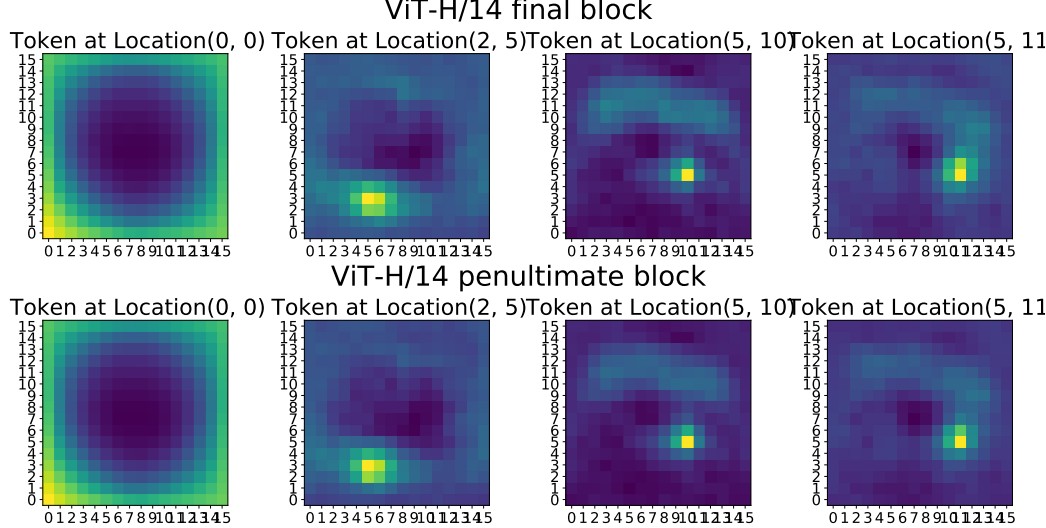

**Figure D.1: Localization heatmaps for ViT-H/14**. We see that ViT-H/14 is also well localized, both in the final block and penultimate block, with tokens with corresponding locations in the interior of the image most similar to the image patches at those locations, while tokens on the edge are similar to many edge positions.

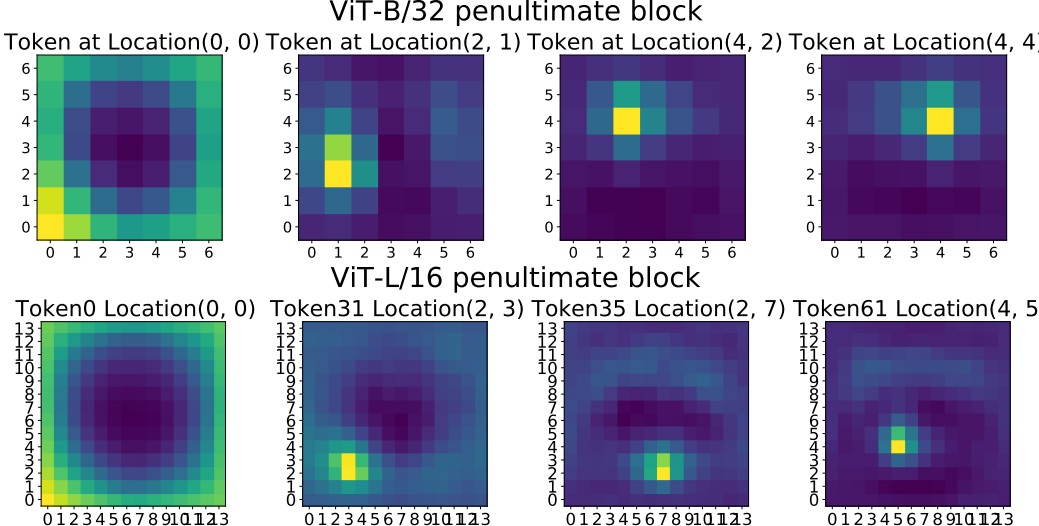

**Figure D.2: Additional localization heatmaps for other higher layers of ViT-L/16 and ViT-B/32**. We see that models (as expected) remain well localized in higher layers other than the final block.

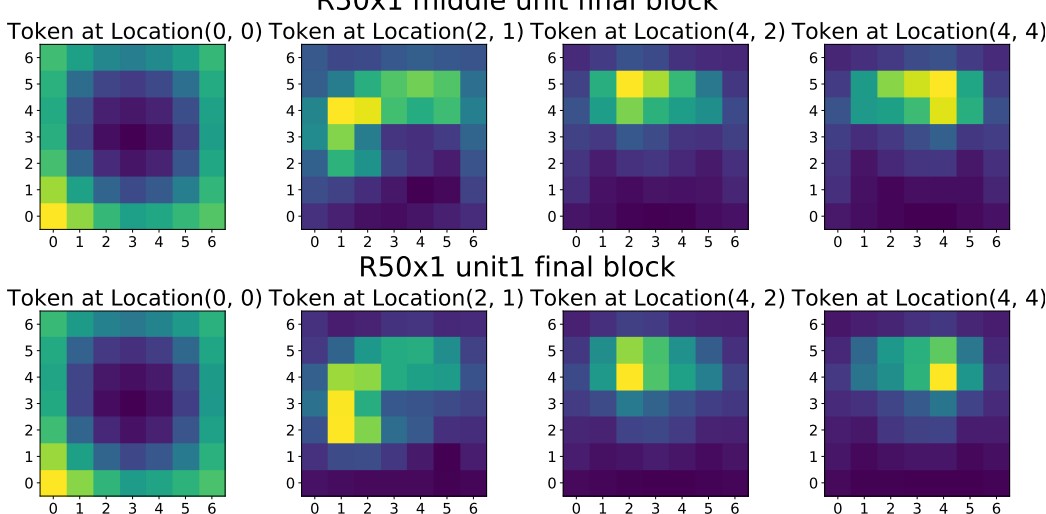

**Figure D.3: Localization heatmaps for layers of ResNet below final layer**. Comparing to Figure 9 in the main text, we see that layers in the ResNet lower than the final layer display better localization, but still not as clear as the CLS trained ViT models.

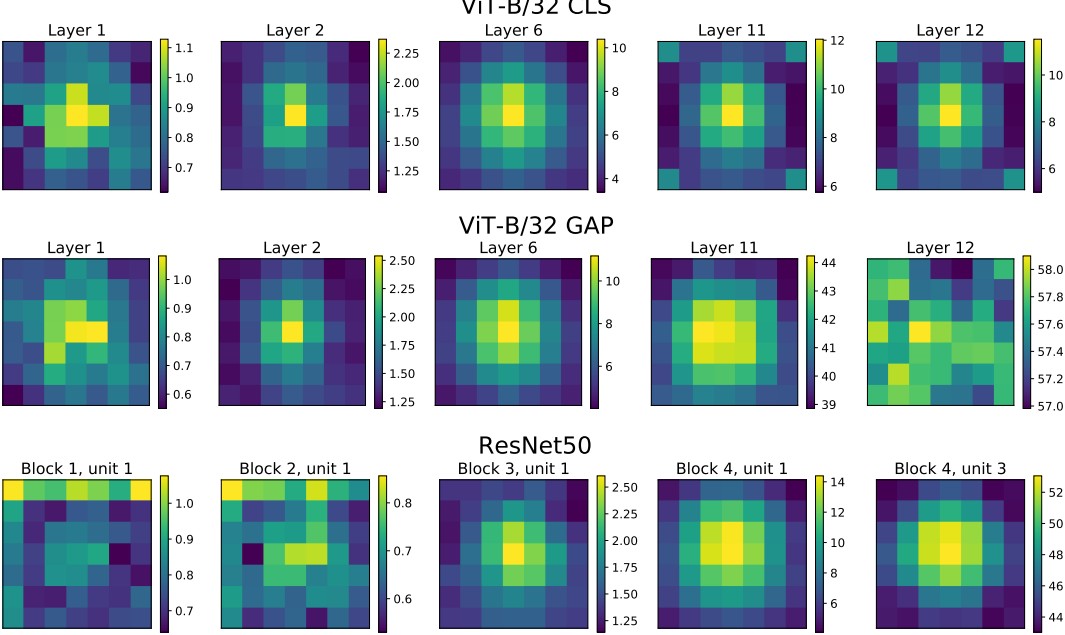

**Figure D.4:** Linear probes spatial localization. We train a linear probe on each individual token and plot the average accuracy over the test set, in percent. Here we plot the results for each token a subset of layers in 3 models: ViT-B/32 trained with a classification token (CLS) or global average pooling (GAP), as well as a ResNet50. Note the different scales of values in different sub-plots.

# E Additional Representation Propagation Results

Figure E.1 shows the ratio of representation norms between skip connections and MLP and Self-Attention Blocks. In both cases, we observe that the CLS token representation is mostly unchanged in the first few layers, while later layers change it rapidly, mostly via MLP blocks. The reverse is true for the spatial tokens representing image patches, whose representation is mostly changed in earlier layers and does not change much during later layers. Looking at the cosine similarity of representations between output in Figure E.2 confirms these findings: while spatial token representations change more in early layers, the output of later blocks is very similar to the representation present on the skip connections, while the inverse is true for the CLS token.

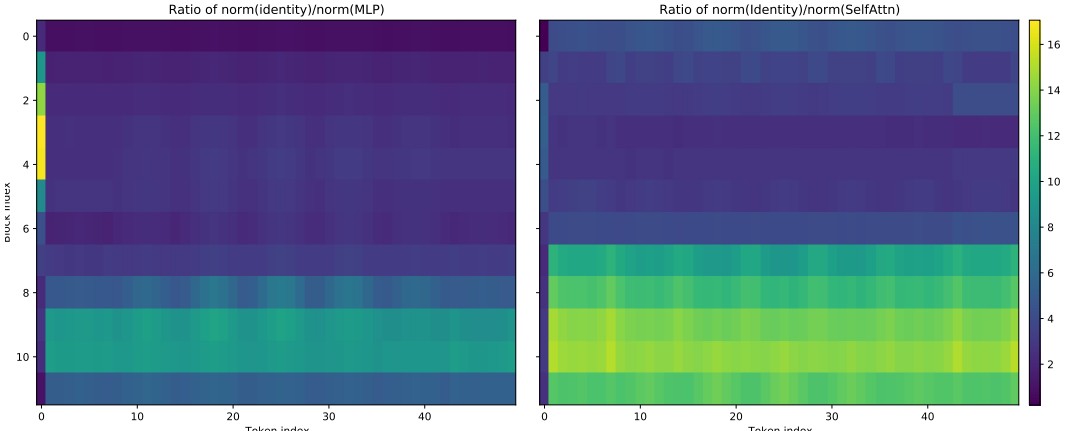

**Figure E.1: Additional Heatmaps of Representation Norms** Ratio Representation Norms $||z_i||/||f(z_i)||$ between skip connection and the MLP or Self-Attention block for the hidden representation of each block on ViT-B/32, separately for each Token (Token 0 is CLS).

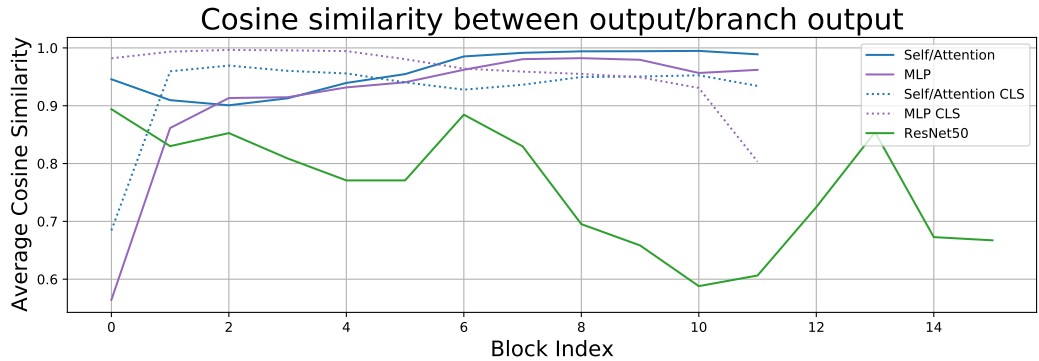

**Figure E.2: Most information in ViT passes through Identity Connections**. Cosine Similarity of representations between the skip-connection (identity) and the longer branch for ViT-B/16 trained on ImageNet and a ResNet v1. For ViT, we show the CLS token separately from the rest of the representation.

# F    Additional results on linear probes

Here we provide additional results on linear probes complementing Figures 11 and 13 of the main paper. In particular, we repeat the linear probes on the CIFAR-10 and CIFAR-100 datasets and, in some cases, add more models to comparisons. For CIFAR-10 and CIFAR-100, we use the first 45000 images of the training set for training and the last 5000 images from the training set for validation. Additional results are shown in Figures F.1, F.2, F.3.

Moreover, we discuss the results in the main paper in more detail. In Figure 11 (left) we experiment with different ways of evaluating a ViT-B/32 model. We vary two aspects: 1) the classifier with which the model was trained, classification token (CLS) or global average pooling (GAP), 2) The way the representation is aggregated: by just taking the first token (which for the CLS models is the CLS token), averaging all tokens, or averaging all tokens except for the first one.

There are three interesting observations to be made. First, CLS and GAP models evaluated with their "native" representation aggregation approach – first token for CLS and GAP for GAP – perform very similarly. Second, the CLS model evaluated with the pooled representation performs on par with the first token evaluation up to last several layers, at which point the performance plateaus. This suggests that the CLS token is crucially contributing to information aggregation in the latter layers. Third, linear probes trained on the first token of a model trained with a GAP classifier perform very poorly for the earlier layers, but substantially improve in the latter layers and almost match the performance of the standard GAP evaluation in the last layer. This suggests all tokens are largely interchangeable in the latter layers of the GAP model.

To better understand the information contained in individual tokens, we trained linear probes on all individual tokens of three models: ViT-B/32 trained with CLS or GAP, as well as ResNet50. Figure 11 (right) plots average performance of these per-token classifiers. There are two main observations to be made. First, in the ViT-CLS model probes trained on individual tokens perform very poorly, confirming that the CLS token plays a crucial role in aggregating global class-relevant information. Second, in ResNet the probes perform poorly in the early layers, but get much better towards the end of the model. This behavior is qualitatively similar to the ViT-GAP model, which is perhaps to be expected, since ResNet is also trained with a GAP classifier.

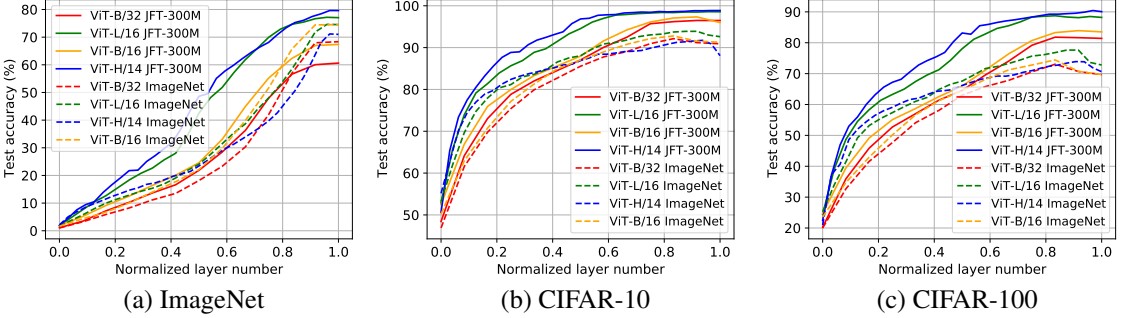

**Figure F.1:** Experiments with linear probes. Additional results on models pre-trained on JFT-300M and ImageNet (Fig. 13 left) – with the addition of ViT-B models and CIFAR-10/100 datasets.

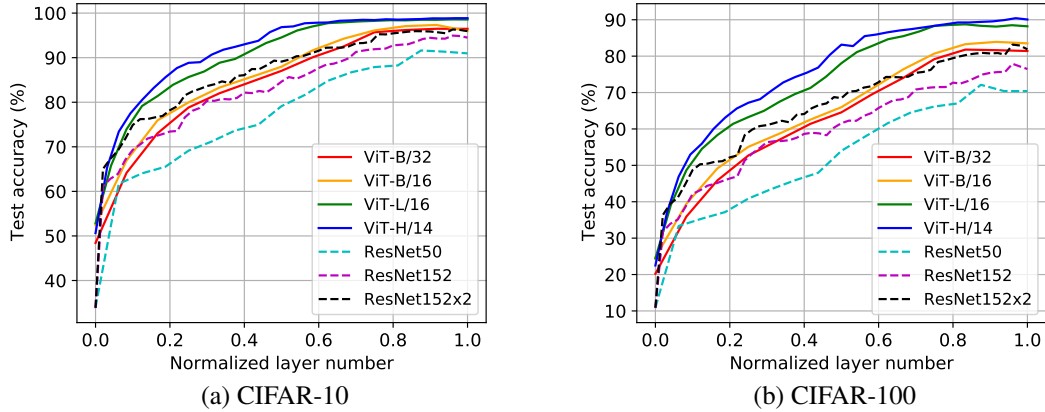

**Figure F.2:** Experiments with linear probes. Additional results on comparison of ViT and ResNet models (Fig. 13 right) on CIFAR-10/100 datasets.

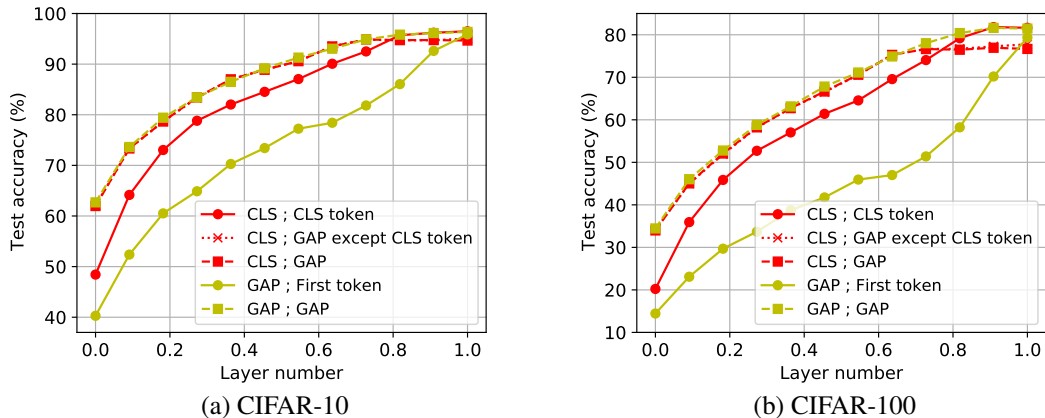

**Figure F.3:** Experiments with linear probes. Additional results on comparison of ViT and ResNet models (Fig. 11 right) on CIFAR-10/100 datasets.

## G   Effects of Scale on Transfer Learning

Finally, we study how much representations change through the finetuning process for a model pre-trained on JFT-300M, finding significant variation depending on the dataset. For tasks like ImageNet or Cifar100 which are very similar to the natural images setting of JFT300M, the representation does not change too much. For medical data (Diabethic Retinopathy detection) or satellite data (RESISC45), the changes are more pronounced. In all cases, it seems like the first four to five layers remain very well preserved, even accross model sizes. This indicates that the features learned there are likely to be fairly task agnostic, as seen in Figure G.1 and Figure G.2.

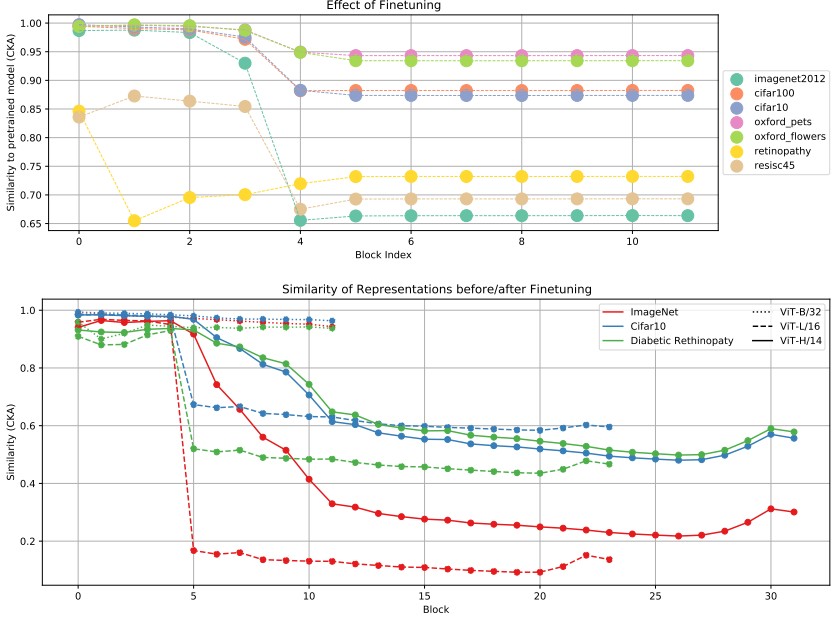

**Figure G.1:** (top) Similarity of representations at each block for ViT-B/16 models compared to before finetuning. (bottom) Similarity of representations at each block for different ViT model sizes.

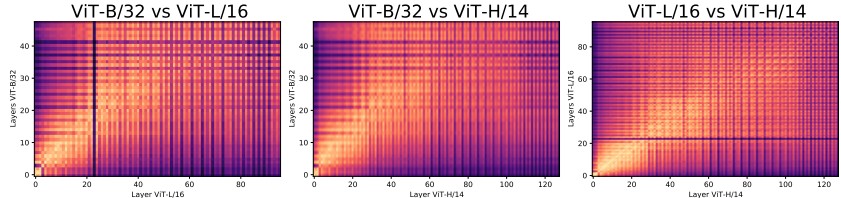

**Figure G.2:** Similarity of representations of different ViT model sizes.

# H   Preliminary Results on MLP-Mixer

Figure H.1 shows the representations from various MLP-Mixer models. The representations seem to also fall very clearly into distinct, uncorrelated blocks, with a smaller block in the beginning and a larger block afterwards. This is independent of model size. Comparing these models with ViT or ResNet as in Figure H.2 models makes it clear that overall, the models behave more similar to ViT than ResNets (c.f. Fig. 1 and 2).

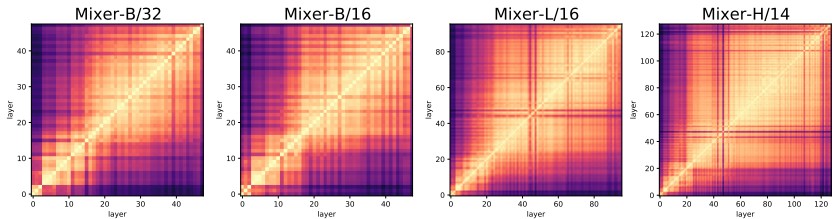

**Figure H.1:** Similarity of representations of different ViT model sizes.

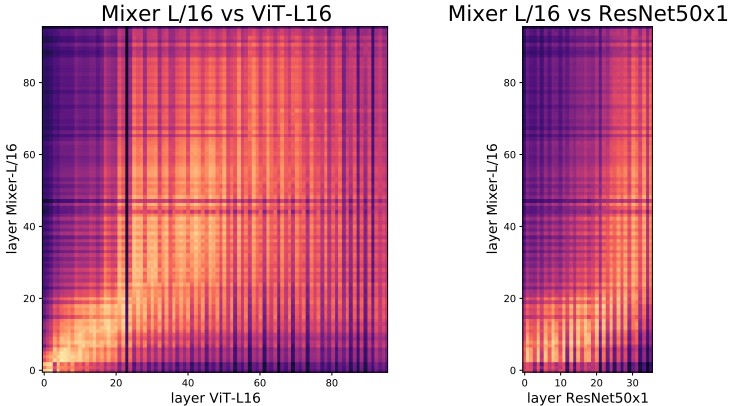

**Figure H.2:** Similarity of representations of different ViT model sizes.