# OpenReview forum: "Do Vision Transformers See Like Convolutional Neural Networks?"
_NeurIPS.cc/2021/Conference — NeurIPS 2021 Poster_

### Official Review · Reviewer_16Eb · 2021-07-13

**Rating:** 4
**Confidence:** 4

**Summary:**

The paper studies the representation of features learned by convnet and transformers. It evaluates the similarity of representations within and between architectures using Centered Kernel Alignment. It studies models trained on datasets of different sizes.

**Limitations And Societal Impact:**

Yes the authors adequately addressed the limitations and potential negative societal impact of their work

**Main Review:**

Strength:
- The study is interesting and quite new
- The paper is well written and easy to follow

Weakness:
- The paper gives insights on the representation of features in convnets and transformers. Nevertheless from a practical point of view it does not allow to understand why this kind of representation emerges and which kind of representation should be privileged in function of the task we wish to solve.

- The results in classification are only with linear probes which is a limited setup

- For the transformers there are only results with ViT architecture and for the convnet there are only results with BiT architecture.
The BiT architecture is not necessarily the most representative of the convnet. It would be interesting to have results with other transformers architecture and more classical convnet architecture to see how the results changes.
"For CNNs, we look at ResNet50x1 which also shows saturating performance when pretraining on JFT-300M, and also ResNet152x2, which in contrast shows large performance gains with increased pretraining dataset size." L416

**Time Spent Reviewing:**

2

---

> ### Author Response · Authors · 2021-08-10
> **Author Response to Reviewer**
>
> Thank you for your review and feedback. We respectfully disagree with the assessment of the paper, and provide specific responses below:
>
> **Representations in ViTs/CNNs:** We are not sure we fully understand the reviewer’s comment on this. In particular, the results of the paper exactly show that the representations that arise in ViT (highly uniform throughout) emerge as a result of global information aggregation by self-attention, which is strongly propagated by the skip connections. We also study many other properties of ViTs and CNNs, from spatial localization, analysis with linear probes and behavior during transfer learning. These experiments demonstrate further key differences between the architectures and how they function.
>
> **Linear probes:** The results using linear probes are another way of analyzing properties of ViTs and CNNs, and not the main way classification is performed --- all models are trained following the original ViT paper. The linear probe experiments show striking results that lower ViT representations are much higher quality (from the lens of classification) than ResNet, and that large datasets are key to these high quality representations (by comparing JFT and ImageNet representations.)
>
> **BiT and CNNs:** We believe BiT is likely to be representative of high performing ConvNets. ViTs need large training set to perform on par with ConvNets, and we choose to study JFT-pretrained models because it is more interesting to compare architectures with similar performance. BiT is a modification of the ResNet architecture, which is a high performing, successful convolutional network, as well as reproducible results with JFT-pretraining. Moreover, the characteristic skip connections (and stages of reduced spatial size) have become ubiquitous in subsequent architectures such as DenseNet and EfficientNet, which is likely to induce similar representational properties. Finally, there are also results in prior work, e.g. (*Similarity of Neural Network Representations Revisited, Kornblith et al*), which show natural correspondences between ResNets and other CNN representations. We will clarify this in the text!

---

> > ### Comment · Reviewer_16Eb · 2021-08-31
> > **Thanks for your response**
> >
> > Thanks for your response, the paper presents an interesting analysis. However the rebuttal do not answer the point on the motivation for the analysis. Indeed, there are no recommendations on how to use these results in practice. The justification for using BiT which is a modified ResNet rather than a classical ResNet and the justification for using only one type of architecture does not seem very convincing.
> > So I keep my initial rating.

---

### Official Review · Reviewer_NaL7 · 2021-07-15

**Rating:** 6
**Confidence:** 5

**Summary:**

This work addresses some fundamental questions on Vision Transformers including 1. how are Vision Transformers solving vision tasks? 2. Do they act like convolutions, learning the same inductive biases from scratch? 3. And what is the role of scale in learning these representations? To answer these questions, they investigate the internal representation structure of ViTs and CNN by using representational similarity techniques like CKA (Centered Kernel Alignment).

**Limitations And Societal Impact:**

1. Some of the properties proposed in this work have been discussed by some previous works [1, 2, 3]. For example, "ViT having more uniform representations, with greater similarity between lower and higher layers" and "ViT incorporates more global
36 information than ResNet at lower layers" have been visualized by the T2T-ViT (Figure1 in T2T-ViT) [2]; and "incorporating local information at lower layers remains vital, with large-scale pre-training data helping early attention layers learn to do this", which is exactly what T2T module [2] aims to solve. This work fails to discuss these.
2. This work provides many interesting properties of ViT but does not solve any problems proposed by its findings.

[1]. Touvron, Hugo, et al. "Going deeper with image transformers." arXiv preprint arXiv:2103.17239 (2021).
[2]. Yuan, Li, et al. "Tokens-to-token vit: Training vision transformers from scratch on imagenet." arXiv:2101.11986 (2021).
[3]. Zhou, Daquan, et al. "Deepvit: Towards deeper vision transformer." arXiv preprint arXiv:2103.11886 (2021).

**Main Review:**

This paper aims to answer some fundamental questions on Vision Transformers that why it can work as well as CNNs. By using CKA as a tool to calculate the representational similarity, it finds many interesting properties of ViT and the final conclusions and results of this work are helpful for the community.

**Time Spent Reviewing:**

4

---

> ### Author Response · Authors · 2021-08-10
> **Author Response to Reviewer**
>
> Thank you very much for the review and helpful pointers! We are glad that you find the results interesting and helpful for the community! Specific responses below:
>
> **Additional references and T2T-ViT:** Thank you for the additional paper references, we will include these in the updated version! Regarding the T2T results on lower layers: we think you might mean Figure 2 in the paper (which is the feature visualization)? This is indeed a relevant (though different) analysis, and we will discuss this in the updated version.
>
> **Properties of ViT vs problems:** The goal of this paper was to provide a better understanding of many of the properties of ViT, but, motivated by some of the discovered differences between ViT and CNNs, we hope to explore hybrid architectures and (informed by the results on localization), new approaches to object detection in the future.

---

### Official Review · Reviewer_MKzA · 2021-07-16

**Rating:** 6
**Confidence:** 4

**Summary:**

This paper deeply studies the working mechanism of vision transformers and explores the similarities and differences between vision transformers and CNNs.

**Limitations And Societal Impact:**

- Despite the comprehensive analysis, there is a lack of providing insights that could help design better network architectures.

- The authors say that they provide ramifications for future applications in Section H but I cannot find exactly where the ramifications
 are.

**Main Review:**

- The originality of this paper is clear. Unlike most previous work that target designing novel network architectures or advanced attention mechanisms, this paper aims to deeply study what the vision transformers learn and help understand the differences between vision transformers and CNNs.

- Through the representation analysis methods, the authors well visualize what the transformer layers learn at different network depth and show how vision transformers learn local and global information.

- The presentation is good and the novelty is significant. I do think the explanations and conclusions can help in designing better vision transformers or hybrid networks.

**Time Spent Reviewing:**

5

---

> ### Author Response · Authors · 2021-08-10
> **Author Response to Reviewer**
>
> Thank you for the review and feedback! We are happy that the results were found to be interesting! Some specific responses below:
>
> **Designing better network architectures:** while the focus of our paper was primarily on understanding ViTs, there are two results that are particularly promising for new architectures. Firstly, the results of Figures 3,4 (and Appendix Figure C.5) which show that learning to attend locally requires a large enough dataset, and may noticeably help performance suggest the use of hybrid CNN + ViT architectures (ViT with initial convolutional layers) in low data regimes. Secondly, the results on spatial localization, which illustrate that ViT (with the CLS token) maintains spatial information very faithfully suggest it might be possible to directly utilize that for object detection applications.
>
> **Section H and future applications:** By this, we refer to Section H in the Appendix, where we also provide some analysis on the concurrently proposed MLP-Mixer architecture, finding that representation structure, which also shows blocks of uniform structure, is more similar to ViT than ResNet

---

> > ### Comment · Reviewer_MKzA · 2021-08-11
> > **Comment after rebuttal**
> >
> > Thanks for the response. After going throught the response and the reviews from other reviewers, I still think the biggest problem of this paper is not clearly explaining why they did such a study. The responses provided by the authors have been identified by previous work, such as the original ViT paper (designing hybrid architecture). The responses do not give attractive points to me. So, I rate the score to 6 regarding the detailed analysis on ViT and CNNs.

---

### Official Review · Reviewer_fR12 · 2021-07-19

**Rating:** 5
**Confidence:** 4

**Summary:**

This paper provides a comprehensive analysis of visual features learned from vision transformers and CNNs. It applies central kernel alignment (CKA) to compare ViT and CNNs from several aspects mainly including a) local and global representations; b) roles of skip connections; and c) spatial localization.

**Limitations And Societal Impact:**

no broader impact section is included in the paper

**Main Review:**

Strengths:
1) ViT and its variants are demonstrating better performance than CNNs in many vision applications, thus understanding the differences between ViTs and CNNs is an important problem.
2) The paper provides thorough analysis and comparisons of ViT and CNN features based on models trained on both small and large datasets.

Weakness:
1) This is basically an analysis paper. While some of its findings are helpful for understanding the differences between the two architectures, the primary claims about the local/global representations and skip connections are not extremely surprising. The paper also doesn't suggest much about what makes ViT superior to CNNs or how to improve the performance of ViT further.

2) I am not quite convinced by the analysis of spatial localization. Why is locality-preserving in the higher layers of ViT so relevant to object detection? How would this property be helpful for object detection if ViT is not learning contextual information? it would be better if the authors could add analysis on how this property would affect ViT in the task of object detection (either in a positive or negative way).

3) Were all the models trained using the same dataset and augmentation in this paper?  It would be good to provide the classification accuracies of these models.  ResNet50 and ResNet152 are not the SOTA CNN models.  How would a better-performed model like EfficientNet affect the analysis?

4) In Fig. 1, both CNN models, especially ResNet50, indicate two blocks along the diagonal. Is there any explanation for this pattern?

5) Do different heads result in different mean attention distances? Some more analysis on this would be interesting.

6) It’s not clear to me how the effective receptive field for VIT is defined and computed.



**Time Spent Reviewing:**

6

---

> ### Author Response · Authors · 2021-08-10
> **Author Response to Reviewer**
>
> Thank you for your review and feedback. We include detailed responses below.
>
> **This is indeed an analysis paper:** with the numerous advances focused on pushing performance, our work instead focuses on better understanding how/why these advances work. We respectfully disagree with the claims being unsurprising. Firstly, the work on analyzing attention and skip connections is only one part of the results (and provide more insight into the representational differences seen in Figure 1), with additional results on localization, transfer learning behavior, and dataset size. Secondly, in our work on analyzing attention, while one might hypothesize that attention would enable incorporation of more global information, the importance of learning to attend locally, and the role of scale (in both datasets and models), was very striking (Figures 3, 4 and Appendix Figure C.5). For the results on the effect of skip connections, prior to this analysis, it was unknown (and difficult to hypothesize) how important skip connections might be in each of these architectures. Overall, before this work (as far as we are aware) there was limited knowledge on the representational structure of ViTs and CNNs and if/what differences there may be between them.
>
> **Spatial Localization Analysis:** ViT preserving spatial information in higher layers is important for object detection as the architecture (and the information it propagates), must enable the determination of an accurate bounding box --- for this, the context present in the image and information on where this context is spatially located are important. Our experiments find that using a CLS token more faithfully preserves this information over global average pooling, and we leave further exploration of this to future work.
>
> **Models and datasets in paper:** As described in Section 3 and Appendix Section A, all models are trained using the on JFT (and ImageNet) as in the original Vision Transformer paper (An Image is worth 16x16 words, Dosovitskiy et al). Specifically, we use the BiT ResNets (from General Visual Representation Learning, Kolelesnikov et al)  used as baselines in the ViT paper. These modified ResNet architectures are comparable or superior to EfficientNet.
>
> **Figure 1 ResNets:** The blocks along the diagonal in the representation heatmap likely result from the different (spatial) stages of the resnet architecture, as noted in the figure text and in prior work, e.g. (*Do Wide and Deep Neural Networks learn the same things, Nguyen et al*). We will emphasize this further!
>
> **Different attention heads:** The different attention heads do indeed result in different mean attention distances, and this is the focus of the results shown in Figures 3, 4 (and Appendix Figure C.5). In Figure 5 we quantitatively analyze the ramifications of this for the representations.
>
> **Effective receptive field of ViT:** In the caption of Figure 6, we briefly describe the procedure. To elaborate, assume we have an image $\mathbf{x}$, and let $\mathbf{y} \in \mathbb{R}^c$ be the representation of the center location in a ResNet feature map or of the center token of a ViT model extracted from a given layer. To compute the effective receptive field for this image, we calculate
> $\mathbf{r}(\mathbf{x}) = \frac{1}{c}\sum_{i=1}^{c} \Bigg|\frac{\partial y_c}{\partial \mathbf{x}}\Bigg|$. To obtain the receptive fields in the paper, we compute effective receptive fields for 32 images and then average them. (This computation is somewhat expensive because it requires a separate backpropagation pass for each image and channel.) Additional details and results are included in Appendix Section C.
>
> **Final note** Thank you for the review and we hope these responses help clarify some of the questions raised. In light of these responses we respectfully ask the reviewer to consider raising their score in support of accepting this paper.

---

> > ### Comment · Reviewer_fR12 · 2021-08-31
> > **Comment after rebuttal**
> >
> > Thanks for the response. After reading the rebuttal and the comments from other reviewers, I still feel
> > the paper didn't provide many helpful insights on designing new architectures or improving existing ones. Further, the
> > claim that "a CLS token more faithfully preserves local information over global average pooling" needs better justifications by experiments. Thus I stand by my original rating.

---

### Decision · Program_Chairs · 2021-09-28

**Decision:**

Accept (Poster)

**Comment:**

After the discussion phase the reviewers scores were borderline with one recommending rejection, one leaning towards reject and two leaning towards accept. All reviewers found the paper to be well written and addresses an important problem of understanding the differences and similarities between ViT and CNN architectures. However, the reviewers all said the work fell short on providing analysis that was both sufficiently thorough (i.e., do these findings extend to different CNN architectures?) and sufficiently insightful, lacking explanations either as to why the observations hold true and/or what actions to take as a result of these findings. Without this, the study is somewhat incomplete. After considering all private comments, reviews, and discussion, the AC does not recommend acceptance and instead encourages the authors to expand their study to verify its validity on other CNN architectures and to further hypothesize and test reasons behind the observations made.

**Consistency Experiment:**

NeurIPS has a long history of experimentation. In 2014, NeurIPS ran an experiment in which 10% of submissions were reviewed by two independent committees to quantify the randomness in the review process. This year, we repeated a variant of this experiment to see how the quality of the review process has changed over time.  This paper was part of the experiment and was therefore assigned to two committees (consisting of reviewers, an Area Chair, and a Senior Area Chair) that reached independent decisions.  If both committees made the same recommendation, this recommendation was followed. If a single committee recommended acceptance, the paper was accepted (with the exception of a few cases in which the other committee identified what we considered a fatal flaw, e.g., an error in a key result).

This copy’s committee reached the following decision: **Reject**

The other committee assigned to the paper recommended **Accept (Spotlight)**.  You can find the other set of reviews, along with any follow up discussion with the authors here:
https://openreview.net/forum?id=R-616EWWKF5